# Risk Factors for the Development of Colistin Resistance during Colistin Treatment of Carbapenem-Resistant *Klebsiella pneumoniae* Infections

Po-Han Huang,[a] Wen-Yin Chen,[b] Sheng-Hua Chou,[c] Fu-Der Wang,[a] Yi-Tsung Lin[a,c]

[a]Division of Infectious Diseases, Department of Medicine, Taipei Veterans General Hospital, Taipei, Taiwan
[b]Division of Infectious Diseases, Department of Paediatrics, Taipei Veterans General Hospital, Taipei, Taiwan
[c]Institute of Emergency and Critical Care Medicine, National Yang Ming Chiao Tung University, Taipei, Taiwan

**ABSTRACT** Colistin is one of the last-resort options for carbapenem-resistant *Klebsiella pneumoniae* (CRKP) infections if novel antibiotics are unavailable, where the development of colistin resistance during treatment represents a major challenge for clinicians. We aimed to investigate the risk factors associated with the development of colistin resistance in patients with CRKP infections following colistin treatment. We conducted a retrospective case-control study of patients with CRKP strains available before and after colistin treatment at a medical center in Taiwan, between October 2016 and November 2020. Cases ($n = 35$) included patients with an initial colistin-susceptible CRKP (ColS-CRKP) strain and a subsequent colistin-resistant CRKP (ColR-CRKP) strain. Controls ($n = 18$) included patients with ColS-CRKP as both the initial and subsequent strains. The 30-day mortality rate after the subsequent CRKP isolation was not different between cases and controls (12/35 [34%] versus 5/18 [28%] [$P = 0.631$]). $bla_{KPC}$ ($n = 38$) and $bla_{OXA-48}$ ($n = 11$) accounted for the major mechanisms of carbapenem resistance. Alterations in *mgrB* were found in 18/35 (51%) ColR-CRKP strains, and *mcr-1* was not detected in any of the strains. More patients received combination therapy in the control group than in the case group (17/18 versus 21/35 [$P = 0.008$]). The logistic regression model indicated that combination therapy with tigecycline was protective against the acquisition of colistin resistance (odds ratio, 0.17; 95% confidence interval, 0.05 to 0.62 [$P = 0.008$]). We observed that the inclusion of tigecycline in colistin treatment mitigated the risk of acquiring colistin resistance. These results offer insight into using the combination of tigecycline and colistin for the treatment of CRKP infections in antimicrobial stewardship.

**IMPORTANCE** Treatment of carbapenem-resistant *Klebsiella pneumoniae* (CRKP) infections is challenging due to the limited options of antibiotics. Colistin is one of the last-resort antibiotics if novel antimicrobial agents are not available. It is crucial to identify modifiable clinical factors associated with the emergence of resistance during colistin treatment. Here, we found that the addition of tigecycline to colistin treatment prevented the acquisition of colistin resistance. Colistin-tigecycline combination therapy is therefore considered a hopeful option in antimicrobial stewardship to treat CRKP infections.

**KEYWORDS** colistin, resistance emergence, protective factors, carbapenem, *Klebsiella pneumoniae*

Address correspondence to Yi-Tsung Lin, ytlin8@vghtpe.gov.tw.

The authors declare no conflict of interest.

Infection by carbapenem-resistant *Klebsiella pneumoniae* (CRKP) causes significant morbidity and mortality, and its global spread poses a major threat to public health (1). Colistin remains an important "last-resort" antibiotic for patients with CRKP infections because novel agents against CRKP are unavailable in many countries (2). However, colistin resistance may develop from the addition of cationic groups (phosphoethanolamine or 4-

amino-4-deoxy-L-arabinose) to lipid A within lipopolysaccharides. Modifications of lipid A are usually mediated by chromosomal mutations of genes encoding the two-component systems PmrAB and PhoPQ, by the inactivation of the *mgrB* gene, or from the plasmid-mediated *mcr-1* gene (3–6). The development of colistin resistance in CRKP makes treatment even more challenging and leads to an increased mortality rate (7, 8).

Several studies have explored the risk factors for colistin resistance during CRKP infections (9–11). Previously, we have shown that colistin exposure is an independent risk factor for the *in vivo* emergence of colistin resistance in CRKP (12). However, studies regarding the risk factors for colistin resistance development after colistin treatment for CRKP infections are lacking. Previous studies have shown conflicting results with regard to the use of combination therapy with colistin, where combination therapy may either suppress the emergence of colistin-resistant *K. pneumoniae* subpopulations or not reduce the emergence of colistin resistance in carbapenem-resistant Gram-negative bacteria (13, 14). However, the above-mentioned studies included a limited number of CRKP isolates.

We conducted this case-control study to identify potential risk factors for colistin resistance development following colistin treatment for CRKP infections as well as to clarify the impacts of different antibiotic combinations on resistance emergence.

## RESULTS

**Study population and clinical characteristics.** During the 49-month study period, we identified 35 (case) and 18 (control) patients from whom a colistin-resistant CRKP (ColR-CRKP) strain and a colistin-susceptible CRKP (ColS-CRKP) strain were isolated, respectively, 7 to 28 days after intravenous colistin treatment for a ColS-CRKP infection. There was no outbreak of CRKP infection identified at our hospital, and there were no more than two patients in the same unit infected by CRKP during the same period. The colistin MICs for the initial and subsequent CRKP strains are shown in Table S1 in the supplemental material. The demographic and clinical characteristics of the case and control groups are shown in Table 1. This study found a high mortality rate for patients with CRKP infections. There were no significant differences between the case and control groups in the in-hospital mortality rate (54% [19/35] versus 44% [8/18] [$P = 0.497$]), 30-day mortality rate after the initial CRKP isolation (23% [8/35] versus 22% [4/18] [$P = 1.000$]), or 30-day mortality rate after the subsequent CRKP isolation (34% [12/35] versus 28% [5/18] [$P = 0.631$]).

The most common CRKP infections among the 53 patients included pneumonia ($n = 27$; 51%), primary bacteremia ($n = 8$; 15%), and urinary tract infection ($n = 5$; 9%). The culture sites of the initial and subsequent CRKP strains are shown in Table S2. The median numbers of days of colistin treatment for cases and controls were 9 days (interquartile range [IQR], 7 to 14 days) and 12 days (IQR, 10 to 13 days), respectively ($P = 0.118$) (Table S3). Twenty-one (60%) case and 17 (94%) control patients received combination therapy ($P = 0.008$). The regimens used in combination therapy are shown in Table 2. The most common combination was colistin plus tigecycline, which was used in 13 (37%) and 14 (78%) patients in the case and control groups, respectively ($P = 0.005$). Ceftazidime-avibactam, meropenem-vaborbactam, and imipenem-cilastatin-relebactam were unavailable at the hospital during the study period.

**Microbiological characteristics of CRKP strains.** Table 3 shows the antimicrobial susceptibilities of the 53 initial CRKP isolates. Twenty-six (84%) case and 14 (78%) control CRKP isolates were susceptible to tigecycline, with no statistical difference being observed ($P = 0.478$). Overall, carbapenemase-producing strains were the majority in both the case and control groups ($n = 52$; 98%) (Table S4). The most common carbapenemase gene was $bla_{KPC}$ ($n = 38$; 72%), followed by $bla_{OXA-48}$ ($n = 11$; 21%), $bla_{NDM}$ ($n = 2$; 4%), and $bla_{IMP}$ ($n = 1$; 2%). K47 was the most common capsular type in the CRKP strains ($n = 37$; 70%), followed by K64 ($n = 8$; 15%) and KN2 ($n = 4$; 8%) (Table S4). No statistical difference was observed between cases and controls in the proportions of carbapenemases and capsular types of the initial CRKP strains.

The quantification of *pmrH* mRNA expression showed that 97% (34/35) of the ColR-

**TABLE 1** Demographic and clinical characteristics of case and control groups[g]

| Variable | Case (n = 35) | Control (n = 18) | P value |
|---|---|---|---|
| | **Value for group** | | |
| Mean age (yrs) (±SD) | 65 (±17) | 66 (±15) | 0.833 |
| No. (%) of male patients | 21 (60) | 15 (83) | 0.085 |
| Median LOS before initial CRKP isolation (days) (IQR) | 27 (12–45) | 13 (1–28) | 0.035 |
| Mean no. of days between initial and subsequent CRKP isolation (±SD) | 17 (±6) | 17 (±5) | 0.650 |
| Median Charlson comorbidity index (IQR) | 8 (5–10) | 8 (5–9) | 0.698 |
| | | | |
| No. (%) of patients with comorbidity | | | |
| Malignancy | 18 (51) | 7 (39) | 0.386 |
| Solid tumor | 15 (43) | 6 (33) | 0.502 |
| Hematological malignancy | 3 (9) | 1 (6) | 1.000 |
| Diabetes mellitus | 18 (51) | 8 (44) | 0.630 |
| Chronic kidney disease[a] | 23 (66) | 11 (61) | 0.741 |
| ESRD[b] | 11 (31) | 2 (11) | 0.177 |
| Heart failure | 8 (23) | 3 (17) | 0.730 |
| Coronary artery disease | 11 (31) | 3 (17) | 0.333 |
| PAOD | 7 (20) | 2 (11) | 0.701 |
| Cirrhosis | 7 (20) | 2 (11) | 0.701 |
| Cerebrovascular accident | 9 (26) | 7 (39) | 0.322 |
| COPD | 3 (9) | 4 (22) | 0.211 |
| Connective tissue disease[c] | 4 (11) | 0 (0) | 0.287 |
| Solid-organ transplantation | 6 (17) | 2 (11) | 0.701 |
| Chronic ventilator dependence[d] | 7 (20) | 3 (17) | 1.000 |
| | | | |
| No. (%) of patients with immunosuppression[e] | 29 (83) | 10 (56) | 0.049 |
| Corticosteroid treatment | 25 (71) | 9 (50) | 0.123 |
| Immunosuppressant treatment | 7 (20) | 3 (17) | 1.000 |
| Chemotherapy | 0 (0) | 1 (6) | 0.340 |
| Neutropenia | 2 (6) | 2 (11) | 0.598 |
| | | | |
| Mean APACHE II score (±SD) | 21 (±6) | 20 (±10) | 0.710 |
| No. (%) of patients with medical device | | | |
| Mechanical ventilator | 28 (80) | 15 (83) | 1.000 |
| Tracheostomy | 7 (20) | 6 (33) | 0.326 |
| Central venous catheter | 29 (81) | 15 (83) | 1.000 |
| Urinary catheter | 28 (80) | 14 (78) | 1.000 |
| Nasogastric or nasojejunal tube | 32 (91) | 17 (94) | 1.000 |
| Surgical drain[f] | 14 (40) | 6 (33) | 0.635 |
| Surgery | 5 (14) | 6 (33) | 0.154 |
| | | | |
| No. (%) of patients with no source control | 1 (3) | 0 (0) | 1.000 |
| Median length of ICU stay (IQR) | 11 (3–17) | 12 (1–15) | 0.539 |
| In-hospital mortality rate [no. (%) of patients] | 19 (54) | 8 (44) | 0.497 |
| 30-day mortality rate after 1st CRKP isolation [no. (%) of patients] | 8 (23) | 4 (22) | 1.000 |
| 30-day mortality rate after 2nd CRKP isolation [no. (%) of patients] | 12 (34) | 5 (28) | 0.631 |

[a]Defined according to KDIGO 2012 clinical practice guidelines for the evaluation and management of chronic kidney disease (31).
[b]Defined as an estimated glomerular filtration rate (GFR) of <15 mL/min/1.73 m$^2$ or dialysis dependence for more than 30 days.
[c]Defined according to the Charlson comorbidity index.
[d]Defined as ventilator dependence for more than 30 days prior to the initial CRKP isolation.
[e]Defined as meeting one of the following criteria: use of corticosteroids or immunosuppressants, receiving chemotherapy, or neutropenia. Use of corticosteroids was defined as receipt of a corticosteroid at a dose of ≥10 mg per day of prednisolone for more than 5 days in the 30 days prior to the subsequent CRKP isolation. Chemotherapy was defined as the receipt of cytotoxic antineoplastic drugs for cancer within 30 days before the subsequent CRKP isolation. Neutropenia was defined as a peripheral absolute neutrophil count of <0.5 × 10$^9$ cells/L.
[f]Defined as the presence of drainage tubes after surgeries or invasive procedures.
[g]CRKP, carbapenem-resistant *Klebsiella pneumoniae*; IQR, interquartile range; LOS, length of hospital stay; ESRD, end-stage renal disease; PAOD, peripheral arterial occlusive disease; COPD, chronic obstructive pulmonary disease; APACHE, Acute Physiology and Chronic Health Evaluation; ICU, intensive care unit.

CRKP strains had higher expression levels than their colistin-susceptible counterparts (Table S1). Only TVGH-CR22 had no significant difference in *pmrH* mRNA expression levels between the ColS-CRKP and ColR-CRKP strains. We quantified another gene of the *pmrHFIJKLM* operon, *pmrK*, in strain TVGH-CR22, and the *pmrK* mRNA expression level of the ColR-CRKP strain was higher than that of its colistin-susceptible counterpart (Table S1). No ColR-CRKP strain carrying the *mcr-1* gene was detected. Alterations in

**TABLE 2** Combination therapy during the interval between the initial and subsequent carbapenem-resistant *K. pneumoniae* strains

| Antibiotic therapy | No. (%) of patients receiving therapy in group | | *P* value |
| | Case (*n* = 35) | Control (*n* = 18) | |
|---|---|---|---|
| Combination therapy | 21 (60) | 17 (94) | 0.008 |
| Colistin + tigecycline | 13 (37) | 14 (78) | 0.005 |
| Colistin + carbapenem[a] | 11 (31) | 9 (50) | 0.187 |
| Colistin + aminoglycoside[b] | 2 (6) | 0 (0) | 0.543 |
| Colistin + tigecycline + carbapenem[a] | 4 (11) | 4 (22) | 0.421 |
| Colistin + aminoglycoside[b] + carbapenem[a] | 1 (3) | 0 (0) | 1.000 |

[a]Includes ertapenem, imipenem-cilastatin, meropenem, and doripenem.
[b]Includes amikacin and gentamicin.

*mgrB* were found in 18 (51%) ColR-CRKP strains, while amino acid substitutions in PmrAB and PhoPQ were detected in 14 (40%) and 2 (6%) ColR-CRKP strains. Among the sequential CRKP isolates of the case group, 30 (86%) initial ColS-CRKP strains had pulsed-field gel electrophoresis (PFGE) patterns (≤3 different bands) similar to those of their subsequent ColR-CRKP counterparts. The PFGE results are shown in Fig. S1.

**Risk factors for the emergence of colistin resistance.** As shown in Tables 1 and 2, patients in the case group were more likely to be male and immunosuppressed. Patients in the case group had also stayed in the hospital for a longer duration before the initial ColS-CRKP isolation, and fewer of them received colistin-tigecycline combination therapy for CRKP infections. Age, sex, and variables with *P* values of <0.1 in the univariate analysis were entered into a stepwise backward selection logistic regression model (Table 4). No use of colistin-tigecycline combination therapy was the only significant risk factor for the development of colistin resistance. Colistin-tigecycline combination therapy was found to be protective against colistin resistance (odds ratio, 0.17; 95% confidence interval, 0.05 to 0.62 [*P* = 0.008]) (Table 4).

## DISCUSSION

In the current study, the inclusion of tigecycline with colistin treatment was found to protect against the development of colistin resistance in patients with CRKP infections treated with colistin. We also found a notably high mortality rate among patients with CRKP strains. Our previous work demonstrated that colistin treatment is an independent risk factor for the *in vivo* emergence of colistin resistance in CRKP (12). Therefore, it is crucial to identify modifiable clinical factors associated with the development of resistance during colistin treatment. The use of combination antibiotic ther-

**TABLE 3** Antimicrobial susceptibilities of 53 initial CRKP strains

| Antibiotic(s) | No. (%) of patients with susceptible initial strain in group | | *P* value |
| | Case (*n* = 35) | Control (*n* = 18) | |
|---|---|---|---|
| Ceftazidime | 0 (0) | 0 (0) | |
| Cefepime | 1 (3) | 1 (6) | 1.000 |
| Piperacillin-tazobactam | 0 (0) | 0 (0) | |
| Ertapenem | 0 (0) | 0 (0) | |
| Imipenem | 0 (0) | 0 (0) | |
| Amikacin | 34 (97) | 16 (89) | 0.263 |
| Gentamicin | 19 (54) | 8 (44) | 0.497 |
| Ciprofloxacin | 0 (0) | 1 (6) | 0.340 |
| Levofloxacin | 0 (0) | 1 (6) | 0.340 |
| Tigecycline | 29 (83) | 13 (72) | 0.478 |
| Trimethoprim-sulfamethoxazole | 6 (17) | 5 (28) | 0.478 |

**TABLE 4** Univariate and multivariate analyses of clinical factors associated with the development of colistin resistance in CRKP strains during colistin treatment[a]

| Variable | Univariate analysis | | Multivariate analysis | |
|---|---|---|---|---|
| | OR (95% CI) | *P* value | OR (95% CI) | *P* value |
| Age | | 0.833 | | |
| Male sex | 0.30 (0.07–1.23) | 0.085 | | |
| LOS before initial CRKP isolation | | 0.035 | | |
| Immunosuppression | 3.87 (1.08–13.90) | 0.049 | | |
| Colistin-tigecycline combination therapy | 0.35 (0.14–0.87) | 0.005 | 0.17 (0.05–0.62) | 0.008 |

[a]CRKP, carbapenem-resistant *Klebsiella pneumoniae*; OR, odds ratio; CI, confidence interval; LOS, length of hospital stay.

apy has been proposed as a potential method to mitigate this development. An *in vitro* study conducted by Deris et al. found that colistin-doripenem combination therapy reduced the emergence of colistin-resistant multidrug-resistant *K. pneumoniae* strains, but only one CRKP isolate was included in this study (13). Clinical studies on the risk factors for the development of colistin-resistant strains during colistin treatment for CRKP infections are limited. Recently, Dickstein et al. conducted a secondary analysis of a randomized controlled trial (RCT) and concluded that colistin-carbapenem combination therapy could not prevent colistin resistance compared to colistin monotherapy in patients infected with carbapenem-resistant organisms; however, only five initial colistin-susceptible isolates were CRKP (14). In line with this secondary analysis, our study did not find a protective effect of colistin-carbapenem combination therapy, but our study found that the inclusion of tigecycline with colistin treatment prevented the development of colistin resistance. While the availability and uptake of novel antimicrobial agents are critical elements in the fight against CRKP, the findings of our study indicate that colistin-tigecycline combination therapy for CRKP infections may be a promising option in antimicrobial stewardship.

In the secondary analysis conducted by Dickstein et al. (14), the authors aimed to compare colistin resistance development following colistin-meropenem combination therapy versus colistin monotherapy in patients infected with carbapenem-resistant organisms. These organisms included several types of carbapenem-resistant Gram-negative bacteria, including *Enterobacterales* (such as *Escherichia coli* and *K. pneumoniae*) and nonfermenting Gram-negative bacilli (such as *Acinetobacter baumannii* and *Pseudomonas aeruginosa*). The study design was robust, as no other antimicrobials targeting Gram-negative bacteria were allowed, and the follow-up isolates in this RCT were regularly obtained through surveillance rectal swabs. Our research was an observational study and gave us the opportunity to compare different antibiotic regimens. With a median duration (10 days) of colistin treatment similar to that reported by Dickstein et al. (14), our real-world observation data corresponded to the notion that the addition of carbapenem to colistin therapy could not prevent resistance development. Our findings also indicate that further RCTs are warranted to validate the role of tigecycline in preventing colistin resistance during colistin treatment of CRKP.

The combination of tigecycline and colistin in the treatment of CRKP infections has been well studied *in vitro* and *in vivo*. Using a time-kill assay, Pournaras et al. found that the colistin-tigecycline combination is synergistic and bactericidal against *K. pneumoniae* carbapenemase (KPC)-producing CRKP isolates (15). Within a mouse model, Fergadaki et al. suggested that treatment with tigecycline, either as monotherapy or in combination with other antibiotics (including colistin and/or meropenem), significantly prolonged survival in KPC-producing CRKP infections (16). Using a time-kill assay, Tian et al. found that the combination of polymyxin B and tigecycline demonstrated bactericidal activity against CRKP strains with heteroresistance to polymyxin B and tigecycline (17). Clinical studies on the effect of colistin-tigecycline combination therapy on CRKP infections are usually conducted before the availability of novel agents against this organism, where better results are reported with regimens containing colistin and

tigecycline (18, 19). Our study was not designed to evaluate the impact of combination therapy on patient outcomes but does indicate a plausible benefit of colistin-tigecycline combination therapy. This potential benefit is particularly desirable in real-world practice, as the absence of routine laboratory testing for colistin resistance prevents physicians from detecting the emergence of resistance in a timely manner.

This study had several limitations. We had a relatively small sample size, and the statistical analysis may not have sufficient power to extrapolate the results to the overall population. However, studies on the collection of paired sequential CRKP isolates to evaluate the development of colistin resistance are limited in the literature. In addition, only patients with CRKP isolates available before and after colistin treatment were included, and the collection of specimen cultures was decided by the treating physician. Patients with CRKP infections treated successfully with colistin did not have subsequent cultures available and were not enrolled as our study controls. Therefore, the population at risk for colistin resistance may be overrepresented. Furthermore, no ColR-CRKP strain carrying the *mcr-1* gene was detected in our study, so whether the findings could be generalizable to ColR-CRKP strains with *mcr-1* acquisition was unknown. Finally, we did not perform whole-genome sequencing for phylogenetic analysis. While the acquisition of colistin resistance in this study was not limited to *in vivo* emergence, and some cases might acquire colistin-resistant strains exogenously, the result is more reflective of real-world practice.

In conclusion, our case-control study demonstrated that combination therapy with tigecycline protects against the development of colistin resistance in patients with CRKP infections receiving colistin treatment. Further prospective interventional studies are needed to validate this association as well as to examine whether this protective effect can be translated into a higher likelihood of clinical success. Our results also offer insight into antimicrobial stewardship, and colistin-tigecycline combination therapy may be a promising strategy to curb colistin resistance when treating CRKP infections.

## MATERIALS AND METHODS

**Study design and patients.** This study was conducted at the Taipei Veterans General Hospital in Taiwan between October 2016 and November 2020. A case-control study was undertaken to identify the risk or protective factors associated with the development of colistin resistance following colistin treatment of CRKP infections. We defined cases as patients from whom an initial ColS-CRKP strain was isolated and, after 7 to 28 days, a ColR-CRKP strain was isolated. We defined controls as patients from whom a ColS-CRKP strain was isolated initially and after 7 to 28 days of treatment. Both case and control groups experienced ≥5 days of intravenous colistin treatment for the initial ColS-CRKP infections. All patients in each group met the study's inclusion criteria. Intravenous colistin was administered at a loading dose of 5 mg/kg of body weight followed by a maintenance dose of 2.5 mg × (1.5 × creatinine clearance + 30) every 12 h. The Institutional Review Board of Taipei Veterans General Hospital approved and waived the informed consent of this study.

**Data collection and definitions.** We collected demographic and clinical data from the electronic chart review, including length of stay before the initial ColS-CRKP isolation, comorbidities, Charlson comorbidity index, immunosuppression (through corticosteroid usage, immunosuppressant usage, receipt of chemotherapy, or neutropenia), and Acute Physiology and Chronic and Prevention Evaluation (APACHE) II score calculated 24 h before or after culture collection. The use of mechanical ventilation, the presence of indwelling medical devices, the length of intensive care unit stay, and antibiotics administered during the interval between the initial and subsequent CRKP strain isolations were also included. Combination therapy was defined as the administration of colistin and other antibiotics commonly used to treat CRKP for >72 h, regardless of the MIC or site of infection. These antibiotics included tigecycline, carbapenem, or aminoglycosides. Source control was defined as the removal of infected medical devices or drainage of infected fluid collections within 7 days after obtaining the initial culture.

**Microbiological study of *K. pneumoniae* strains.** *Klebsiella pneumoniae* was identified using matrix-assisted laser desorption ionization–time of flight mass spectrometry (bioMérieux). We determined the MICs of antibiotics other than colistin and tigecycline with the Vitek-2 system (bioMérieux), and the results were interpreted according to Clinical and Laboratory Standards Institute criteria (20). In this study, CRKP was defined as a *K. pneumoniae* isolate that was nonsusceptible (MIC ≥ 2 mg/L) to imipenem or meropenem. Colistin resistance was defined as an MIC of >2 mg/L according to European Committee on Antimicrobial Susceptibility Testing (EUCAST) 2022 version 12.0 guidelines (http://www.eucast.org/clinical_breakpoints), as determined by broth microdilution. Tigecycline susceptibility was based on U.S. Food and Drug Administration criteria (susceptible, MIC ≤ 2 mg/L; resistant, MIC ≥ 8 mg/L), as determined by an Etest (bioMérieux) (21).

As previously described, we used *wzi* sequencing to determine the capsular types of the CRKP strains

(22, 23). All CRKP isolates were screened for carbapenemase genes, including $bla_{KPC}$, $bla_{OXA-48}$, $bla_{IMP}$, and $bla_{NDM}$, as previously described (24). To determine the colistin resistance mechanism, we identified the presence of the *mcr-1* gene and alterations in the *mgrB*, *phoPQ*, *pmrAB*, and *crrAB* genes as previously described (4, 25). These genes were then amplified by PCR, and the nucleotides were determined by Sanger sequencing. Subsequently, we compared the gene sequences in ColR-CRKP strains to those in their ColS-CRKP counterparts. We then compared the amino acid alignments with Clustal Omega and identified the insertion sequences using ISfinder as previously described (12, 26, 27). Moreover, we evaluated the mRNA expression levels of the *pmrH*, *pmrK*, and 16S rRNA genes using real-time quantitative reverse transcription-PCR as previously described (3, 4, 12). The relative expression of target genes in CRKP strains was compared to that in a colistin-susceptible strain, NTUH-K2044 (expression = 1; colistin MIC = 1 mg/L). The $\Delta\Delta C_T$ method was used with normalization to 16S rRNA levels for analysis. Primers for PCR are listed in Table S5 in the supplemental material.

To evaluate the genetic relatedness of the initial colistin-susceptible and subsequent colistin-resistant strains in the case group, we conducted PFGE as previously described (28, 29). The results were interpreted according to Tenover criteria, with paired strains being considered genetically indistinguishable or closely related if they had no more than three band differences (30).

**Statistical analyses.** Categorical variables were compared using the chi-square test or Fisher's exact test, as appropriate. Continuous variables were analyzed using Student's *t* test or a Mann-Whitney U test, as appropriate. Multivariate analysis was run for age, sex, and all variables with *P* values of <0.1 in univariate analyses using a stepwise backward selection logistic regression model. All statistical analyses were performed using Statistical Package for the Social Sciences software version 23.0 (SPSS, Chicago, IL, USA). For all analyses, we considered a two-tailed *P* value of <0.05 to be statistically significant.

## SUPPLEMENTAL MATERIAL

Supplemental material is available online only.

**SUPPLEMENTAL FILE 1**, PDF file, 0.6 MB.

## ACKNOWLEDGMENTS

We thank the Medical Science and Technology Building of Taipei Veterans General Hospital for providing experimental space and facilities.

Y.-T.L. participated in the study design. P.-H.H. and W.-Y.C. participated in data collection and statistical analysis. S.-H.C. and Y.-T.L. participated in the laboratory experiment. P.-H.H. and Y.-T.L. participated in data analysis and drafted the manuscript. All authors took responsibility for the accuracy of the data analysis. All authors read and approved the final manuscript.

We declare no conflict of interest.

This work was supported by grants from the Ministry of Science and Technology in Taiwan (MOST 108-2314-B-010-030-MY3), Taipei Veterans General Hospital (V111A-008, V110C-068, and V111C-061), the National Taiwan University Hospital-Taipei Veterans General Hospital Joint Research Program (VN 108-02), and the Szu-Yuan Research Foundation of Internal Medicine. The funders had no role in the study design, data collection and interpretation, or the decision to submit the work for publication.

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
