## [Reviewer comments · Microbiology Spectrum]

Microbiology Spectrum

Risk factors for the development of colistin resistance during colistin treatment of carbapenem-resistant *Klebsiella pneumoniae* infections

Po-Han Huang, Wen-Yin Chen, Sheng-Hua Chou, Fu-Der Wang, and Yi-Tsung Lin

Corresponding Author(s): Yi-Tsung Lin, Taipei Veterans General Hospital

Review Timeline:

Submission Date:	January 29, 2022
Editorial Decision:	March 8, 2022
Revision Received:	May 16, 2022
Accepted:	May 18, 2022

Editor: Rosemary She

Reviewer(s): Disclosure of reviewer identity is with reference to reviewer comments included in decision letter(s). The following individuals involved in review of your submission have agreed to reveal their identity: Melese Hailu Legese (Reviewer #1); Sharmi Naha (Reviewer #2); Le Van Chuong (Reviewer #4)

Transaction Report:

DOI: <https://doi.org/10.1128/spectrum.00381-22>

March 8, 2022

Dr. Yi-Tsung Lin
Taipei Veterans General Hospital
Division of Infectious Diseases, Department of Medicine
Number 201, Section 2, Shih-Pai Road, Beitou District, Taipei City, Taiwan 11217
Taipei
Taiwan

Re: Spectrum00381-22 (Risk factors for the development of colistin resistance during colistin treatment of carbapenem-resistant *Klebsiella pneumoniae* infections)

Dear Dr. Yi-Tsung Lin:

Thank you for submitting your manuscript to Microbiology Spectrum. Your manuscript will be reconsidered if you address the reviewer comments. Please note that some of the reviewer comments are provided in the form of an attachment to this email. When submitting the revised version of your paper, please provide (1) point-by-point responses to the issues raised by the reviewers as file type "Response to Reviewers," not in your cover letter, and (2) a PDF file that indicates the changes from the original submission (by highlighting or underlining the changes) as file type "Marked Up Manuscript - For Review Only". Please use this link to submit your revised manuscript - we strongly recommend that you submit your paper within the next 60 days or reach out to me. Detailed instructions on submitting your revised paper are below.

Link Not Available

Sincerely,

Rosemary She

Journals Department
Reviewer comments:

Reviewer #1 (Comments for the Author):

Dear author,

Please get all my comments and suggestions in the file attached.

Reviewer #2 (Comments for the Author):

Authors are required to modify their manuscript as per comments. This will enhance the quality of their work and would add substantial knowledge to the subject

Reviewer #3 (Comments for the Author):

Po-Han Huang and colleagues collected colistin-resistant and colistin-susceptible CRKP isolates before and after the patients receiving colistin treatment. This study described the basic clinical and microbiological information and performed the robust statistical analysis. The results of this study could be biased due to the small sample size and simple analysis without control.

Major comments,

1. Line 69, 135-138. The sample size is also limited in this study, which could lead to bias of the results from statistical analysis. I think the authors should detailly discuss the potential bias of the results, but not just simply describe that it is limited (line 169-170).
2. Line 78. How to define epidemiological linking? Please supply the definition and how to define the linking among the patients in this study.
3. Line 82. What control groups were used in the comparison of this study? I suppose the "high mortality" should refer to other groups.
4. Line 87-89. It seems that the CRKP isolates before and after treatment could be from different body sites for patients, which leaves uncertainty that the colistin-resistant CRKP could not be evolved from the colistin-susceptible CRKP. There are chances that the isolates collected after colistin treatment could be from a newly acquired infection, which is also confirmed by the PFGE result in this study. Thus, it could be biased to conclude the development of ColR-CRKP from ColS-CRKP in this study.
5. Line 98-105. The authors simply described the microbiological characteristics of collected CRKP isolates, which seems not associated with the main topic of this study. Please rewrite the results and describe the logic of these results associated with the development of ColR.
6. Line 106-109. The author only detects some mutations in mgrB genes. However, they did not discuss whether these mutations lead to the ColR, and no other mechanisms of ColR were testified to explain the reason why these ColS-CRKP isolates evolved into ColR.
7. Line 99-109. It would be better to compare the differences in these characteristics between the two groups.
8. In line 83, the authors found no significance between case and control groups, which is not consistent with the description in line 123.

Minor comments,

1. Line 59. The development of colistin resistance needs to add the acquisition of the mcr-1 gene.
2. Line 147-148. I don't understand the definition of nature observational study. The cited references are also real-world observational studies.
3. Please supply the figure of PFGE result.
4. Line 218. What are the MIC ranges of tested antimicrobials? There are some results showed ≥ 64 and 128.
5. Line 222. What version of EUCAST was used?
6. The format of references should be checked carefully.

Reviewer #4 (Comments for the Author):

The authors conducted a case-control study to identify potential risk factors for colistin resistance developing following colistin treatment for carbapenem resistant *K. pneumoniae*. The study has elaborate design, reliable data, taken in quite long period of time (49 months), and many techniques has been applied to clarify the research question. Nevertheless, I have some following comments:

1. The major finding of this paper is that combination therapy with tigecycline was protective against the development of colistin resistance. However, in a study, Van Duin et al. found that 46% of the cases with carbapenem resistant *K. pneumoniae* were also tigecycline-intermediate or -resistant (1). Furthermore, tigecycline is contraindicated in children less than 8 years of age and can not use to treat systemic infections. A recent meta-analysis found that tigecycline-treated patients (especially those treated for ventilator-associated pneumonia) had a higher mortality rate than those taking other antibiotics, leading to a notice from the FDA.
2. The data in line 87, 88 are not correlate to the supplementary table S2.
3. What is the correlation of initial and subsequent strain? Are they the same strain or not? The subsequent strain may come from environment, thus cannot conclude that they are development of colistin resistance during treatment.
4. The method for detection of mcr gene that was applied in this study has the ability to detect other variant of mcr gene or not?
5. The detection of mgrB gene is not cover the most common mechanism resistant to colistin, the author should add phoPQ-PmrAB...
6. The major results are the risk factors for the development of colistin resistance, however too little factors were assumed as

significant factors. The title should be change and the conclusion is not consistency to the research question.

7. The sample size is small, it may affect to the reliable of statistical analysis.

8. The mortality rates in both study group and control group are high, how about the mortality rate in the patients who received prescription colistin + carbapenem group to compare with colistin + tigecycline group?

Minor things: the et al. should be in italic

Staff Comments:

Preparing Revision Guidelines

Please return the manuscript within 60 days; if you cannot complete the modification within this time period, please contact me. If you do not wish to modify the manuscript and prefer to submit it to another journal, please notify me of your decision immediately so that the manuscript may be formally withdrawn from consideration by Microbiology Spectrum.

General comments and suggestions

The authors aimed to assess the “Risk factors for the development of colistin resistance during colistin treatment of carbapenem-resistant *Klebsiella pneumoniae* infections” which is interesting. The findings are relevant and can be utilized for effective antimicrobial stewardship. However, the paper has major limitations and of course the authors acknowledged as the paper had several limitations. On the other hand, the authors still have a chance to advance the data by performing whole genome sequencing which could back up the retrospective nature and other methodological limitations. As shown below, the whole genome data can be utilized to generate more data on these colistin resistant *K. pneumoniae*. I recommend the authors to address those relevant comments.

Major comments

1. Even if the initial plan was risk factors assessment for the development of colistin resistance, all colistin resistance and susceptible carbapenems resistance *K. pneumoniae* could be whole genome sequenced so that very relevant information shown easily. From the whole genome sequence
 - The phylogenetic population of colistin strains could be drawn
 - The sequence type (STs) of *K. pneumoniae* could be shown
 - The different colistin and carbapenems resistance gene variants could be identified. Eg. Variants of *bla*_{NDM} could be identified
 - It is also possible to show whether the colistin and carbapenemase genes are carried in the plasmid or inserted in the chromosome
 - Possible resistance genes for tetracyclines groups that could affect the efficacy of tigecycline could be identified. This could help to see if the colistin resistance strains were susceptible to tigecycline because they don't carried

such resistance genes. The colistin-tigecycline combination treatment could be challenged if tigecycline resistance genes are carried.

2. Investigation of combination treatment for colistin resistance strains looks insufficient
 - a. As the authors said, no investigation done for approved combined antibiotics like ceftazidime-avibactam, meropenem-vaborbactam, and imipenem-cilastatin-relebactam. This could be done easily once the strains are collected.
 - b. While table 3 showed better susceptibility for amikacin (97%) than tigecycline (83%), colistin-amikacin combination efficacy is not investigated. Of course, it was shown in table 2 as “colistin-aminoglycoside” combination but that showed a disagreement with the higher sensitivity of amikacin and the case was fewer in number. The retrospective nature of the study may affected this.
3. The sample size is very limited that affected the conclusion.

Minor comments

Abstract section

- The sentences used to describe the result and conclusion parts are confusing and needs modification
- The none detection of *mcr* genes could be shown in the abstract section
- Information written in the result and conclusion of the abstract section are the same with the section importance. Better to remove the importance section.

Result sections

- The result sections are fine
- Use the term “sensitive” instead of “susceptibly” once the AST pattern is described.

Eg. For table 3

Discussion section

- The discussion section needs further modification.
- Elaboration for the none detection of *mcr* gene is needed.

Methodology section:

- Overall, it is fine
- But it is better to show the different primers used instead of referring readers to go and look for other papers

Throughout the paper,

- There are resistance genes that should be written using italic and subscript form eg in the abstract section KPC, OXA that should be as *bla*_{KPC}, *bla*_{OXA}
- Bacterial names should be written scientifically e.g in the introduction section *Klebsiella pneumoniae* should be written as *Klebsiella pneumoniae*
- The phrase “...the development of colistin resistance...” could be modified as “...the treatment of colistin resistance strains...” as necessary. Eg.
 - Line 33 and 34 “The logistic regression model indicated that combination therapy with tigecycline was protective against the development of colistin resistance...”
 - Line 46 and 47 “We observed that the inclusion of tigecycline in colistin treatment mitigated the development of colistin resistance in CRKP strains. It is better to write it

Comments from Reviewer #1:

General comments and suggestions

The authors aimed to assess the “Risk factors for the development of colistin resistance during colistin treatment of carbapenem-resistant *Klebsiella pneumoniae* infections” which is interesting. The findings are relevant and can be utilized for effective antimicrobial stewardship. However, the paper has major limitations and of course the authors acknowledged as the paper had several limitations. On the other hand, the authors still have a chance to advance the data by performing whole genome sequencing which could back up the retrospective nature and other methodological limitations. As shown below, the whole genome data can be utilized to generate more data on these colistin resistant *K. pneumoniae*. I recommend the authors to address those relevant comments.

Answer: Thank you for the comments. I have made major modifications and I hope the current revision can meet the criteria of publication.

Major comments

1. Even if the initial plan was risk factors assessment for the development of colistin resistance, all colistin resistance and susceptible carbapenems resistance *K. pneumoniae* could be whole genome sequenced so that very relevant information shown easily. From the whole genome sequence
 - The phylogenetic population of colistin strains could be drawn
 - The sequence type (STs) of *K. pneumoniae* could be shown
 - The different colistin and carbapenems resistance gene variants could be identified. Eg. Variants of *bla_{NDM}* could be identified
 - It is also possible to show whether the colistin and carbapenemase genes are carried in the plasmid or inserted in the chromosome
 - Possible resistance genes for tetracyclines groups that could affect the efficacy of tigecycline could be identified. This could help to see if the colistin resistance strains were susceptible to tigecycline because they don't carried such resistance genes. The colistin-tigecycline combination treatment could be challenged if tigecycline resistance genes are carried.

Answer: Thanks for your comments. We agreed with the benefits whole genome sequencing could bring to this study. This study focused on the clinical aspects of resistance development and the possible association with antibiotic use, so whole genome sequencing was not planned as a part of this study. We addressed this issue in the limitation. Please find Page 10, Lines 187-190.

2. Investigation of combination treatment for colistin resistance strains looks insufficient
 - a. As the authors said, no investigation done for approved combined antibiotics like ceftazidime-avibactam, meropenem-vaborbactam, and imipenem-cilastatin-relebactam. This could be done easily once the strains are collected.

Answer: Thanks for your comment. As described in our manuscript, ceftazidime-avibactam, meropenem-vaborbactam, and imipenem-cilastatin-relebactam were unavailable at the hospital during the study period. Please find Page 5, Lines 94-95. We believe it adds to the importance of our study: as novel agents are still

unavailable in many countries, colistin remains an important “last-resort” antibiotic and it is crucial to find a strategy to mitigate the risk of resistance development.

- b. While table 3 showed better susceptibility for amikacin (97%) than tigecycline (83%), colistin-amikacin combination efficacy is not investigated. Of course, it was shown in table 2 as “colistin-aminoglycoside” combination but that showed a disagreement with the higher sensitivity of amikacin and the case was fewer in number. The retrospective nature of the study may affected this.

Answer: Thanks for your comment. As you have mentioned, we were unable to know the exact reason why a certain antibiotic combination was chosen more frequently than another because of the retrospective nature of this study. However, we believe that colistin and aminoglycoside were seldomly used together because both of them are associated with nephrotoxicity, and the high risk of renal failure precluded this combination.

3. The sample size is very limited that affected the conclusion.

Answer: Thanks for your comment. We had a small sample size, but as far as we know this is a large collection of paired CRKP clinical isolates in a study designed to evaluate development of colistin resistance to date; therefore, we believe our study still makes a significant contribution to the literature. We addressed this issue in the limitation. Please find Page 9-10, Lines 178-181.

Minor comments

Abstract section

- The sentences used to describe the result and conclusion parts are confusing and needs modification

Answer: Thanks for your comment. We have revised several sentences in the result and conclusion parts and we hope they offer more clarity now.

- The none detection of *mcr* genes could be shown in the abstract section

Answer: Thanks for your comment. We have added the relevant information in the abstract section as suggested. Please see Page 2, Line 31-32.

- Information written in the result and conclusion of the abstract section are the same with the section importance. Better to remove the importance section.

Answer: Thanks for your comment. The journal required authors to provide the importance section for every submitted article. We have revised the importance section and made the content not so similar to the result and conclusion of the abstract section. Please see Page 3, Line 39-46.

Result sections

- The result sections are fine
- Use the term “sensitive” instead of “susceptibly” once the AST pattern is

described. Eg. For table 3

Answer: Thanks for your comment. We found that the guidelines for antibiotic susceptibility testing (AST) referenced by our work, including European Committee on Antimicrobial Susceptibility Testing (1), Clinical Laboratory and Standards Institute (2), and U.S. Food and Drug Administration (3) criteria, used “susceptible” instead of “sensitive” to describe the AST results. We also noted that a research article about CRKP by Cienfuegos-Gallet et al. (4) recently published at *Microbiology Spectrum* also used “susceptible” to report the AST results. We therefore suppose that using this term will not cause confusion.

Reference

1. Clinical and Laboratory Standards Institute. 2021. Performance standards for antimicrobial susceptibility testing, 31st ed. Clinical and Laboratory Standards Institute, Wayne, PA.
2. The European Committee on Antimicrobial Susceptibility Testing. 2022. Breakpoint tables for interpretation of MICs and zone diameters, version 12.0
3. Wyeth Pharmaceuticals Inc. 2010. Full prescribing information for Tygacil. U.S. Food and Drug Administration. https://www.accessdata.fda.gov/drugsatfda_docs/label/2010/021821s021b1.pdf. Retrieved 16 March 2022.
4. Cienfuegos-Gallet AV, Zhou Y, Ai W, Kreiswirth BN, Yu F, Chen L. 2022. Multicenter Genomic Analysis of Carbapenem-Resistant *Klebsiella pneumoniae* from Bacteremia in China. *Microbiol Spectr.* 1:e0229021.

Discussion section

- The discussion section needs further modification.

Answer: Thanks for your comment. We have revised some of the sentences in the Discussion section and we hope they offer more clarity now. Please see Page 8, Line 146-147; Page 9, Line 156-158; Page 9, Line 178-179; and Page 10, Line 185-190.

- Elaboration for the none detection of *mcr* gene is needed.

Answer: Thanks for your comment. We have added some discussion regarding the non-detection of *mcr-1* gene as suggested and stated that one of the limitations of our study is that whether the findings were generalizable to ColR-CRKP strains with *mcr-1* acquisition was unknown. Please find Page 10, Line 185-187.

Methodology section:

- Overall, it is fine
- But it is better to show the different primers used instead of referring readers to go and look for other papers

Answer: Thanks for your comment. We showed the different primers used in the Supplementary Table S5. We also referred to it in the Materials and Methods section. Please find Page 13, Line 250.

Throughout the paper,

- There are resistance genes that should be written using italic and subscript form eg in the abstract section KPC, OXA that should be as *bla*KPC, *bla*OXA

Answer: Thanks for your comments. We have made the correction. Please see Page 2, Line 30.

- Bacterial names should be written scientifically e.g in the introduction section *Klebsiella pneumoniae* should be written as *Klebsiella pneumoniae*

Answer: Thanks for your comments. We have made the correction. Please see Page 3, Line 40.

- The phrase "...the development of colistin resistance..." could be modified as "...the treatment of colistin resistance strains..." as necessary. Eg.
 - ◆ Line 33 and 34 "The logistic regression model indicated that combination therapy with tigecycline was protective against the development of colistin resistance..."

Answer: Thanks for your comments. We have revised this sentence and replaced "development" with "acquisition". This will add clarity to our presentation. Please see Page 2, Line 33-35.

- ◆ Line 46 and 47 "We observed that the inclusion of tigecycline in colistin treatment mitigated the development of colistin resistance in CRKP strains. It is better to write it

Answer: Thanks for your comment. We have revised the importance section and avoided the use of "...the development of colistin resistance..." in the current version. Please see Page 3, Line 39-46.

Besides, we still used the phrase "development of colistin resistance" to describe the acquisition of colistin resistance in some parts of our manuscript because we found previous studies also presented this way even when the resistance did not emerge from the index colistin-susceptible strain. For example, in the study by Dickstein et al. titled "Colistin Resistance Development Following Colistin-Meropenem Combination Therapy Versus Colistin Monotherapy in Patients With Infections Caused by Carbapenem-Resistant Organisms", the subsequent colistin-resistant strain isolated after colistin treatment could be a different species from the initial colistin-susceptible strain (1). In order to show the relevance to the previous literature, we used the same way of presentation.

Reference

1. Dickstein Y, Lellouche J, Schwartz D, Nutman A, Rakovitsky N, Dishon Benattar Y, Altunin S, Bernardo M, Iossa D, Durante-Mangoni E, Antoniadou A, Skiada A, Deliolanis I, Daikos GL, Daitch V, Yahav D, Leibovici L, Rognås V, Friberg LE, Mouton JW, Paul M, Carmeli Y; AIDA Study Group. 2020. Colistin Resistance Development Following Colistin-Meropenem Combination Therapy Versus Colistin Monotherapy in Patients with Infections Caused by Carbapenem-Resistant Organisms. *Clin Infect Dis* 71:2599-607.

Comments from Reviewer #2:

Authors have assessed the effect of colistin treatment in carbapenem-resistant *Klebsiella pneumoniae*. With rapid spread of carbapenem-resistant *K. pneumoniae*, and decreased affectivity of recommended drugs, use of colistin was adopted. But resistance towards colistin is also increasing, especially in *K. pneumoniae*, due to occurrence of chromosomal resistance. So, studies highlighting outcomes of colistin use in treating carbapenem-resistant *K. pneumoniae* are extremely important to understand their role as savior in this antibiotic crisis situation. The authors have evaluated effect of colistin treatment on carbapenem-resistant *K. pneumoniae* which is highly appreciated. This study holds a promising outcome, despite certain deficiencies which has been described below-

Major comments-

Line 20: Colistin is not the major therapy for CRKP treatment. Other WHO recommended drug combinations are generally adopted unless there is additional resistance to the drugs. Colistin are generally used as the last option. Hence this line needs modification.

Answer: Thanks for your comment. We have revised this sentence and replaced “major therapy” with “one of the last-resort options”. Please see Page 2, Line 20.

Line 47-49: Repetition of line in abstract and importance sections. A different presentation of the work or outcome is expected in “importance section”.

Answer: Thanks for your comment. We have revised the importance section and made the content not so similar to the abstract section. Please see Page 3, Line 39-46.

Line 82: CRKP infections increases mortality. Mortality between the case and control group was found to be comparable in this study. Mortality in case group might be attributed to CRKP infections, but what were the reason for mortality in control group? Explain.

Answer: Thanks for your comment. In this study cases included patients from whom an initial colistin-susceptible CRKP (ColS-CRKP) strain was isolated and, after 7–28 days, a colistin-resistant CRKP (ColR-CRKP) strain was isolated. Controls included patients from whom a ColS-CRKP strain was isolated initially and after 7–28 days of treatment. Therefore, both case and control groups included patients with CRKP infections, which could explain the high mortality rate in both groups. Please find our original description in Page 11, Line 203-205.

Line 95-96: Table 2 showed the combinations of antibiotic used and Table 3 the susceptibility. In Table S3, details of antibiotic usage during strain collection has been given. Susceptibility towards aminoglycoside was more than 40% in both case and control groups. Also tigecycline susceptibility was high. Although tigecycline was used in some patients, why was aminoglycoside not used? Use of colistin should be the last option when no other antibiotics are working. Why in all combinations, colistin was used? It is understood that it's a retrospective study and decision of treatment was out of authors' scope. But a logical explanation should be provided for choosing colistin in all combination therapy. Are any other combination was tried before, upon whose failure, colistin was adopted?

Answer: Thanks for your comment. As you have mentioned, we were unable to know the exact reason why a certain antibiotic combination was chosen more frequently than another because of the retrospective nature of this study. However, we believe that colistin and aminoglycoside were seldomly used together because both of them are associated with nephrotoxicity, and the high risk of renal failure precluded this combination.

Our study aimed to evaluate possible risk factors associated with the development of colistin resistance following colistin treatment for CRKP infections, so the use of colistin, no matter whether it was used alone or in combination with other agents, is the inclusion criteria. Both case and control groups experienced ≥ 5 days of intravenous colistin treatment for the initial ColS-CRKP infections, which was the reason why colistin was used in all combinations in our study. Please find our inclusion criteria in Page 11, Line 205-207, and study aim in Page 4, Line 69-71.

Line 103: When carbapenem-resistant genes (KPC and OXA-48) are present and the isolates were susceptible to aminoglycosides and perhaps tigecycline, then, as obvious theory, combination of tigecycline with aminoglycoside would have been the first choice. Then again why colistin???

Answer: Thanks for your comment. Our study aimed to evaluate possible risk factors associated with the development of colistin resistance following colistin treatment for CRKP infections, so the use of colistin, no matter whether it was used alone or in combination with other agents, is the inclusion criteria. All patients received colistin as the backbone therapy according to our study design.

For the role of colistin in CRKP, we provided the evidence for your reference. A review on treatment of infections caused by carbapenemase-producing *Enterobacteriales* by Rodríguez-Baño et al. pointed out that polymyxins have been a cornerstone in the management of CRE infections when novel agents were not available (1). A study assessing the importance of combination therapy in treating bloodstream infections caused by KPC-producing *K. pneumoniae* by Tumbarello et al. noted that colistin was the most commonly used antibiotic in both monotherapy and combination therapy, and the study concluded that the combination including colistin, tigecycline, and meropenem was associated with lower mortality (2). A pharmacoepidemiology study conducted by Strich et al. showed a notable decrease in colistin prescriptions after the introduction of ceftazidime-avibactam to 210 US hospitals, and it stated that colistin, the antibiotic most commonly included in the antimicrobial regimens, was often the backbone of combination therapy previously (3). As you have mentioned, we were unable to know the exact reason why a certain antibiotic combination was chosen more frequently than another because of the retrospective nature of this study. However, we believe that colistin used to be part of the therapy for CRKP infections at our hospital due to the recommendation and clinical evidence found in the past literature.

Reference

1. Rodríguez-Baño J, Gutiérrez-Gutiérrez B, Machuca I, Pascual A. 2018. Treatment of Infections Caused by Extended-Spectrum-Beta-Lactamase-, AmpC-, and Carbapenemase-Producing *Enterobacteriaceae*. *Clin Microbiol Rev*.

- 14;31(2):e00079-17.
2. Tumbarello M, Viale P, Viscoli C, Trecarichi EM, Tumietto F, Marchese A, Spanu T, Ambretti S, Ginocchio F, Cristini F, Losito AR, Tedeschi S, Cauda R, Bassetti M. 2012. Predictors of mortality in bloodstream infections caused by *Klebsiella pneumoniae* carbapenemase-producing *K. pneumoniae*: importance of combination therapy. *Clin Infect Dis.* 55(7):943-50.
 3. Strich JR, Ricotta E, Warner S, Lai YL, Demirkale CY, Hohmann SF, Rhee C, Klompas M, Palmore T, Powers JH, Dekker JP, Adjemian J, Matsouaka R, Woods CW, Danner RL, Kadri SS. 2021. Pharmacoepidemiology of Ceftazidime-Avibactam Use: A Retrospective Cohort Analysis of 210 US Hospitals. *Clin Infect Dis.* 72(4):611-621.

Line 106: What antibiotic combination was used in isolates with mutation in *mgrB*? Also, details of mutations are lacking. Since, this work is about colistin, details of *mgrB* would have added information to the existing knowledge. Also, mutations of other chromosomal genes responsible for lipopolysaccharide modification (at least *phoPQ*, *pmrAB*) would have been studied too.

Answer: Thanks for your comments. The antibiotic combinations used in isolates with or without mutation in *mgrB* were shown as below. No differences in antibiotic combinations were found between isolates with or without mutation in *mgrB*.

Antibiotics	With mgrB mutation (n = 18)	Without mgrB mutation (n = 17)	P value
Combination therapy			
Colistin + tigecycline	9 (50)	4 (24)	0.105
Colistin + carbapenem ^a	4 (22)	7 (41)	0.227
Colistin + aminoglycoside ^b	1 (6)	1 (6)	1.000
Colistin + tigecycline + carbapenem ^a	2 (11)	2 (12)	1.000
Colistin + aminoglycoside ^b + carbapenem ^a	0 (0)	1 (6)	0.486

Data are expressed as No. (%)

Abbreviations: CRKP, carbapenem-resistant *Klebsiella pneumoniae*

a. Includes ertapenem, imipenem-cilastatin, meropenem, and doripenem.

b. Includes amikacin and gentamicin.

We showed the details of mutated *mgrB* and mutations of other chromosomal genes (including *phoPQ*, *pmrAB*, and *crrAB*) responsible for lipopolysaccharide modification in the Supplementary Table S1. We referred to it in the Result section; please see Page 6, Line 113-115. We also referred to it in the Materials and Methods section; please see Page 12, Line 239-241. We also noted two isolates mistakenly identified as having no mutations in *mgrB*, so the number of isolates with *mgrB* mutations was corrected in the revised manuscript (18/35 [51%]). Please see Page 2, Line 31 and Page 6, Line 113.

Minor comments

Line 40: Italicize *K. pneumoniae*.

Answer: Thanks for your comment. We have made the correction. Please see Page 3, Line 40.

Line 137: Tigecycline is again a last resort drug, and resistance towards it is still limited. But often use of such antibiotic would also lead to resistance. In order to halt this, again other combinations should have been be tried. At least give some insights regarding this issue in the discussion part.

Answer: Thanks for your comment. We have added some relevant discussion on this issue in the discussion part, emphasizing that the availability and uptake of novel antimicrobial agents are critical in the fight against rising antibiotic resistance. Please find Page 8, Line 146-148.

Comments from Reviewer #3:

Po-Han Huang and colleagues collected colistin-resistant and colistin-susceptible CRKP isolates before and after the patients receiving colistin treatment. This study described the basic clinical and microbiological information and performed the robust statistical analysis. The results of this study could be biased due to the small sample size and simple analysis without control.

Major comments,

1. Line 69, 135-138. The sample size is also limited in this study, which could lead to bias of the results from statistical analysis. I think the authors should detailly discuss the potential bias of the results, but not just simply describe that it is limited (line 169-170).

Answer: Thanks for your comment. As we have described in Page 8-9, Lines 151-154, the study by Dickstein et al. drew its conclusion from a collection of various carbapenem-resistant colistin-susceptible organisms, including *A. baumannii* (14/22), *K. pneumoniae* (5/22), and *E. coli* (2/22) (1). In contrast, all clinical isolates included in our study were CRKP strains; therefore, despite the limited number of our samples, we believe our study still contributes to the literature by presenting findings specific for CRKP.

However, we agreed that a small sample size could introduce bias because observations may have a higher risk of being chance findings. Statistical analysis with small sample size is also less likely to have sufficient power to extrapolate the results to the overall population. We have added the relevant explanation into the discussion section. Please see Page 9-10, Line 178-181.

Reference

1. Dickstein Y, Lellouche J, Schwartz D, Nutman A, Rakovitsky N, Dishon Benattar Y, Altunin S, Bernardo M, Iossa D, Durante-Mangoni E, Antoniadou A, Skiada A, Deliolanis I, Daikos GL, Daitch V, Yahav D, Leibovici L, Rognås V, Friberg LE, Mouton JW, Paul M, Carmeli Y; AIDA Study Group. 2020. Colistin Resistance Development Following Colistin-Meropenem Combination Therapy Versus Colistin Monotherapy in Patients with Infections Caused by Carbapenem-Resistant Organisms. *Clin Infect Dis* 71:2599-607.

2. Line 78. How to define epidemiological linking? Please supply the definition and how to define the linking among the patients in this study.

Answer: Thanks for your comment. There was no outbreak of CRKP infection identified at our hospital, and there were no more than two patients in the same unit infected by CRKP during the same period. We have revised this sentence to provide a better understanding of the study population. Please find Page 5, Line 77-78.

3. Line 82. What control groups were used in the comparison of this study? I suppose the "high mortality" should refer to other groups.

Answer: Thanks for your comment. We found that patients infected with CRKP in our study had a high mortality rate compared to that previously reported in the literature. In a prospective, multicenter, cohort study (CRACKLE-2), unadjusted 30-day

mortality of hospitalized patients with cultures positive for CRKP recruited from 71 hospitals in Argentina, Australia, Chile, China, Colombia, Lebanon, Singapore, and the USA was 19% (95% CI 15–22; 93 of 502) (1).

Reference

1. Wang M, Earley M, Chen L, Hanson BM, Yu Y, Liu Z, Salcedo S, Cober E, Li L, Kanj SS, Gao H, Munita JM, Ordoñez K, Weston G, Satlin MJ, Valderrama-Beltrán SL, Marimuthu K, Stryjewski ME, Komarow L, Luterbach C, Marshall SH, Rudin SD, Manca C, Paterson DL, Reyes J, Villegas MV, Evans S, Hill C, Arias R, Baum K, Fries BC, Doi Y, Patel R, Kreiswirth BN, Bonomo RA, Chambers HF, Fowler VG Jr, Arias CA, van Duin D; Multi-Drug Resistant Organism Network Investigators. 2022. Clinical outcomes and bacterial characteristics of carbapenem-resistant *Klebsiella pneumoniae* complex among patients from different global regions (CRACKLE-2): a prospective, multicentre, cohort study. *Lancet Infect Dis.* 22(3):401-412.

4. Line 87-89. It seems that the CRKP isolates before and after treatment could be from different body sites for patients, which leaves uncertainty that the colistin-resistant CRKP could not be evolved from the colistin-susceptible CRKP. There are chances that the isolates collected after colistin treatment could be from a newly acquired infection, which is also confirmed by the PFGE result in this study. Thus, it could be biased to conclude the development of ColR-CRKP from ColS-CRKP in this study.

Answer: Thanks for your comment. As we have mentioned as one of the limitations of our study, the acquisition of colistin resistance was not limited to *in vivo* emergence and some cases might have acquired the subsequent resistant strains from inter-patient transmission or the environment. However, the result may be more reflective of real-world practice. Please see Page 10, Line 187-190.

We used the phrase “development of colistin resistance” to describe the acquisition of colistin resistance because we found previous studies usually presented this way even when the resistance did not emerge from the index colistin-susceptible strain. For example, in the study by Dickstein et al. titled “Colistin Resistance Development Following Colistin-Meropenem Combination Therapy Versus Colistin Monotherapy in Patients With Infections Caused by Carbapenem-Resistant Organisms”, the subsequent colistin-resistant strain isolated after colistin treatment could be a different species from the initial colistin-susceptible strain (1). In order to show the relevance to the previous literature, we used the same way of presentation.

Reference

1. Dickstein Y, Lellouche J, Schwartz D, Nutman A, Rakovitsky N, Dishon Benattar Y, Altunin S, Bernardo M, Iossa D, Durante-Mangoni E, Antoniadou A, Skiada A, Deliolanis I, Daikos GL, Daitch V, Yahav D, Leibovici L, Rognàs V, Friberg LE, Mouton JW, Paul M, Carmeli Y; AIDA Study Group. 2020. Colistin Resistance Development Following Colistin-Meropenem Combination Therapy Versus Colistin Monotherapy in Patients with Infections Caused by Carbapenem-Resistant Organisms. *Clin Infect Dis* 71:2599-607.

5. Line 98-105. The authors simply described the microbiological characteristics of collected CRKP isolates, which seems not associated with the main topic of this study. Please rewrite the results and describe the logic of these results associated with the development of ColR.

Answer: Thanks for your comment. The microbiological characteristics described in our study may not be associated with the main topic, but we believe it is important to provide a background knowledge about the epidemiology of CRKP at our study site. As pointed out in the study by Wang et al. (1), global CRKP epidemics have significant regional differences in bacterial characteristics, and the background microbiological information help readers better interpret our results.

Reference

1. Wang M, Earley M, Chen L, Hanson BM, Yu Y, Liu Z, Salcedo S, Cober E, Li L, Kanj SS, Gao H, Munita JM, Ordoñez K, Weston G, Satlin MJ, Valderrama-Beltrán SL, Marimuthu K, Stryjewski ME, Komarow L, Luterbach C, Marshall SH, Rudin SD, Manca C, Paterson DL, Reyes J, Villegas MV, Evans S, Hill C, Arias R, Baum K, Fries BC, Doi Y, Patel R, Kreiswirth BN, Bonomo RA, Chambers HF, Fowler VG Jr, Arias CA, van Duin D; Multi-Drug Resistant Organism Network Investigators. 2022. Clinical outcomes and bacterial characteristics of carbapenem-resistant *Klebsiella pneumoniae* complex among patients from different global regions (CRACKLE-2): a prospective, multicentre, cohort study. *Lancet Infect Dis.* 22(3):401-412.

6. Line 106-109. The author only detects some mutations in *mgrB* genes. However, they did not discuss whether these mutations lead to the ColR, and no other mechanisms of ColR were testified to explain the reason why these ColS-CRKP isolates evolved into ColR.

Answer: Thanks for your comment. We compared *pmrH* and *pmrK* mRNA expressions of the initial ColS-CRKP and subsequent ColR-CRKP strains, and the significant elevation of mRNA expressions in all CRKP strains demonstrate that the mutations identified led to colistin resistance. We showed the result in Supplementary Table S1. We also showed the details of mutations of other chromosomal genes (including *phoPQ*, *pmrAB*, and *crrAB*) responsible for colistin resistance in the Supplementary Table S1. Please find Page 6, Line 107-112 and Page 12-13, Line 239-250.

7. Line 99-109. It would be better to compare the differences in these characteristics between the two groups.

Answer: Thanks for your comment. We found no differences in the proportion of carbapenemases and capsular types of the initial CRKP strains between case and control groups. Please see Supplementary Table S4 and Page 6, Line 104-106.

8. In line 83, the authors found no significance between case and control groups, which is not consistent with the description in line 123.

Answer: Thanks for your comment. We have revised this sentence and deleted “ColR-” in this sentence. Please see Page 8, Line 131.

Minor comments,

1. Line 59. The development of colistin resistance needs to add the acquisition of the *mcr-1* gene.

Answer: Thanks for your comment. We have revised this sentence and added *mcr-1* gene as a potential mechanism contributing to colistin resistance. Please see Page 4, Line 56-58.

2. Line 147-148. I don't understand the definition of nature observational study. The cited references are also real-world observational studies.

Answer: Thanks for your comment. The original sentence was intended to emphasize that our research is an observational study, which was different from the one by Dickstein et al. which analyzed data from a clinical trial. We have revised this sentence and deleted "in nature" to prevent misunderstanding. Please see Page 9, Line 156-157.

3. Please supply the figure of PFGE result.

Answer: Thanks for your comment. We showed the PFGE result in Supplementary Figure F1. We also mentioned in the Result section. Please see Page 6, Line 117.

4. Line 218. What are the MIC ranges of tested antimicrobials? There are some results showed ≥ 64 and 128.

Answer: Thanks for your comment. The colistin MIC range in our study was from < 0.125 to ≥ 64 . We have revised the data in Supplementary Table S1 and replaced three "128" with " ≥ 64 ".

5. Line 222. What version of EUCAST was used?

Answer: Thanks for your comment. We used the 2022, Version 12.0 EUCAST breakpoint table for bacteria. We have added the relevant information into our manuscript. Please see Page 12, Line 233.

6. The format of references should be checked carefully.

Answer: Thanks for your comment. We have checked the format of references and made corrections as appropriate. Although not required by the *Microbiology Spectrum*, we found that the abbreviate journal names were in italic in recently published articles, so we italicized the journal names in our reference list.

Comments from Reviewer #4:

The authors conducted a case-control study to identify potential risk factors for colistin resistance developing following colistin treatment for carbapenem resistant *K. pneumoniae*. The study has elaborate design, reliable data, taken in quite long period of time (49 months), and many techniques has been applied to clarify the research question. Nevertheless, I have some following comments:

1. The major finding of this paper is that combination therapy with tigecycline was protective against the development of colistin resistance. However, in a study, Van Duin et al. found that 46% of the cases with carbapenem resistant *K. pneumoniae* were also tigecycline-intermediate or -resistant (1). Furthermore, tigecycline is contraindicated in children less than 8 years of age and can not use to treat systemic infections. A recent meta-analysis found that tigecycline-treated patients (especially those treated for ventilator-associated pneumonia) had a higher mortality rate than those taking other antibiotics, leading to a notice from the FDA.

Answer: Thanks for your comment. The MIC breakpoints used in the study by Van Duin et al. was defined by the European Committee on Antimicrobial Susceptibility Testing (EUCAST) at that time, with susceptible, intermediate, and resistant defined as MIC < 2 µg/mL, 2 µg/mL, and > 2 µg/mL, respectively (1). However, the most current version of EUCAST breakpoint table for bacteria (2022, Version 12.0) did not include tigecycline MIC breakpoints for *Enterobacterales* except for *Escherichia coli* and *Citrobacter koseri* (http://www.eucast.org/clinical_breakpoints). We therefore used the interpretive criteria proposed by U.S. Food and Drug Administration, with susceptible, intermediate, and resistant defined as MIC ≤ 2 µg/mL, 4 µg/mL, and ≥ 8 µg/mL, respectively (2).

We are aware that tigecycline should not be used during tooth development (last half of pregnancy, infants, and children less than 8 years of age) unless there are no better alternatives (2). All the patients included in our study were more than 8 years of age. We are also aware that a warning from FDA mentioned an increase in all-cause mortality across 13 phase 3 and 4 trials in tigecycline-treated patients (2), and we agreed the potential harm of tigecycline limits its widespread use. However, there are limited options for treatment of CRKP infections if the novel agent against CRKP is not available, and we believe tigecycline is still one of the “last-resort” choices against CRKP.

Reference

1. van Duin D, Cober E, Richter SS, Perez F, Kalayjian RC, Salata RA, Evans S, Fowler VG, Bonomo RA, Kaye KS. 2015. Residence in Skilled Nursing Facilities Is Associated with Tigecycline Nonsusceptibility in Carbapenem-Resistant *Klebsiella pneumoniae*. *Infect Control Hosp Epidemiol.* 36(8):942-8.
2. Wyeth Pharmaceuticals Inc. 2010. Full prescribing information for Tygacil. U.S. Food and Drug Administration. https://www.accessdata.fda.gov/drugsatfda_docs/label/2010/021821s021lbl.pdf. Retrieved 29 March 2022.

2. The data in line 87, 88 are not correlate to the supplementary table S2.

Answer: Thanks for your comment. We described the focus of CRKP infection in Page 5, Line 86-87, and we described the sites where we obtained the CRKP cultures in Supplementary table S2. They were not always the same; for example, the infection focus was “pneumonia” but the site of CRKP isolation might be “blood” in a patient with bacteremic pneumonia. By contrast, if the site of CRKP isolation was “blood” and there was no other focus of infection identified in a patient, we considered the patient having “primary bacteremia”.

3. What is the correlation of initial and subsequent strain? Are they the same strain or not? The subsequent strain may come from environment, thus cannot conclude that they are development of colistin resistance during treatment.

Answer: Thanks for your comment. Most of the initial and subsequent strains (30/35, 86 %) were similar according to the results from PFGE. The acquisition of colistin resistance was not limited to *in vivo* emergence and some cases might have acquired the subsequent infection from inter-patient transmission or the environment. We have described this issue as the limitation of this study. Please see Page 10, Line 187-190.

We used the phrase “development of colistin resistance” to describe the acquisition of colistin resistance because we found previous studies also presented this way even when the resistance did not emerge from the index colistin-susceptible strain. For example, in the study by Dickstein et al. titled “Colistin Resistance Development Following Colistin-Meropenem Combination Therapy Versus Colistin Monotherapy in Patients With Infections Caused by Carbapenem-Resistant Organisms”, the subsequent colistin-resistant strain isolated after colistin treatment could be a different species from the initial colistin-susceptible strain (1). In order to show the relevance to the previous literature, we used the same way of presentation.

Reference

1. Dickstein Y, Lellouche J, Schwartz D, Nutman A, Rakovitsky N, Dishon Benattar Y, Altunin S, Bernardo M, Iossa D, Durante-Mangoni E, Antoniadou A, Skiada A, Deliolanis I, Daikos GL, Daitch V, Yahav D, Leibovici L, Rognàs V, Friberg LE, Mouton JW, Paul M, Carmeli Y; AIDA Study Group. 2020. Colistin Resistance Development Following Colistin-Meropenem Combination Therapy Versus Colistin Monotherapy in Patients with Infections Caused by Carbapenem-Resistant Organisms. *Clin Infect Dis* 71:2599-607.

4. The method for detection of *mcr* gene that was applied in this study has the ability to detect other variant of *mcr* gene or not?

Answer: Thanks for your comment. The method for detection of *mcr* gene in our study was unable to detect other variants. We have made the corrections and replaced “*mcr*” with “*mcr-1*” throughout the manuscript. Please find Page 2, Line 32; Page 4, Line 58; Page 6, Line 112; Page 10, Line 185-187; and Page 12, Line 240.

5. The detection of *mgrB* gene is not cover the most common mechanism resistant to colistin, the author should add *phoPQ-PmrAB*...

Answer: Thanks for your comment. We showed the details of mutations of other chromosomal genes (including *phoPQ*, *pmrAB*, and *crrAB*) responsible for colistin

resistance in the Supplementary Table S1. We referred to it in the Result section; please see Page 6, Line 113-115. We also referred to it in the Materials and Methods section; please see Page 12, Line 239-245.

6. The major results are the risk factors for the development of colistin resistance, however too little factors were assumed as significant factors. The title should be change and the conclusion is not consistency to the research question.

Answer: Thanks for your comment. As we have described in the original manuscript, we evaluated a wide range of potential risk factors including demographic information, length of stay before the initial CoIS-CRKP isolation, comorbidities, Charlson comorbidity index, immunosuppression (through corticosteroid usage, immunosuppressant usage, receipt of chemotherapy, or neutropenia), and Acute Physiology and Chronic and Prevention Evaluation (APACHE) II score calculated 24 h before or after culture collection. We also assessed the use of mechanical ventilation, presence of indwelling medical devices, length of intensive care unit stay, antibiotics administered during the interval between the initial and subsequent CRKP strain isolation, and source control. Please see Page 11-12, Line 213-224. Age, sex, and variables with P values < 0.1 in the univariate analysis were entered into a stepwise backward selection logistic regression model, and only the use of colistin-tigecycline combination therapy was identified as an independent protective factor against the development of colistin resistance. In other words, no use of the colistin-tigecycline combination therapy was the only significant risk factor of the development of colistin resistance. Please see Page 7, Line 124-126.

7. The sample size is small, it may affect to the reliable of statistical analysis.

Answer: Thanks for your comment. We agreed that a small sample size could introduce bias because observations may have a higher risk of being chance findings. Statistical analysis with small sample size is also less likely to have sufficient power to extrapolate the results to the overall population. We have added some discussion on the limitation of small sample size in the discussion section. Please see Page 9-10, Line 178-181.

8. The mortality rates in both study group and control group are high, how about the mortality rate in the patients who received prescription colistin + carbapenem group to compare with colistin + tigecycline group?

Answer: Thanks for your comment. The 30-day mortality rate was 25.0% (5/20) and 33.3% (9/27) in patients who received colistin-carbapenem and colistin-tigecycline combinations, respectively. The inclusion and exclusion criteria of our study were designed to evaluate risk factors of colistin resistance development, and we do not aim to study the impact of different treatment on mortality.

Minor things: the et al. should be in italic

Answer: Thanks for your comment. We have reviewed the articles recently published by *Microbiology Spectrum* and found that “et al.” were not written in italic. We think we would like to comply with the format and style of this journal.

van Duin D, Cober E, Richter S, et al. Residence in skilled nursing facilities is associated with tigecycline non-susceptibility in carbapenem-resistant *Klebsiella pneumoniae*. *Infect Control Hosp Epidemiol* 2015;36:942-8.

May 18, 2022

Dr. Yi-Tsung Lin
Taipei Veterans General Hospital
Division of Infectious Diseases, Department of Medicine
Number 201, Section 2, Shih-Pai Road, Beitou District, Taipei City, Taiwan 11217
Taipei
Taiwan

Re: Spectrum00381-22R1 (Risk factors for the development of colistin resistance during colistin treatment of carbapenem-resistant *Klebsiella pneumoniae* infections)

Dear Dr. Yi-Tsung Lin:

Your manuscript has been accepted, and I am forwarding it to the ASM Journals Department for publication. You will be notified when your proofs are ready to be viewed.

Sincerely,

Rosemary She
Editor, Microbiology Spectrum
